# LOGICAL ENTITY REPRESENTATION IN KNOWLEDGE-GRAPHS FOR DIFFERENTIABLE RULE LEARNING

**Chi Han**[1]**, Qizheng He**[1]**, Charles Yu**[1]**, Xinya Du**[2]**, Hanghang Tong**[1]**, Heng Ji**[1]
[1]University of Illinois Urbana-Champaign, [2]The University of Texas at Dallas
`{chihan3, qizheng6, ctyu2, htong, hengji}@illinois.edu`
`xinya.du@utdallas.edu`

## ABSTRACT

Probabilistic logical rule learning has shown great strength in logical rule mining and knowledge graph completion. It learns logical rules to predict missing edges by reasoning on existing edges in the knowledge graph. However, previous efforts have largely been limited to only modeling chain-like Horn clauses such as $R_1(x, z) \land R_2(z, y) \Rightarrow H(x, y)$. This formulation overlooks additional contextual information from neighboring sub-graphs of entity variables $x$, $y$ and $z$. Intuitively, there is a large gap here, as local sub-graphs have been found to provide important information for knowledge graph completion. Inspired by these observations, we propose *Logical Entity RePresentation (LERP)* to encode contextual information of entities in the knowledge graph. A LERP is designed as a vector of probabilistic logical functions on the entity's neighboring sub-graph. It is an interpretable representation while allowing for differentiable optimization. We can then incorporate LERP into probabilistic logical rule learning to learn more expressive rules. Empirical results demonstrate that with LERP, our model outperforms other rule learning methods in knowledge graph completion and is comparable or even superior to state-of-the-art black-box methods. Moreover, we find that our model can discover a more expressive family of logical rules. LERP can also be further combined with embedding learning methods like TransE to make it more interpretable. [1]

## 1 INTRODUCTION

In recent years, the use of logical formulation has become prominent in knowledge graph (KG) reasoning and completion (Teru et al., 2020; Campero et al., 2018; Payani & Fekri, 2019), mainly because a logical formulation can be used to enforce strong prior knowledge on the reasoning process. In particular, probabilistic logical rule learning methods (Sadeghian et al., 2019; Yang et al., 2017) have shown further desirable properties including efficient differentiable optimization and explainable logical reasoning process. These properties are particularly beneficial for KGs since KGs are often large in size, and modifying KGs has social impacts so rationales are preferred by human readers.

Due to the large search space of logical rules, recent efforts Sadeghian et al. (2019); Yang et al. (2017); Payani & Fekri (2019) focus on learning *chain-like Horn clauses* of the following form:

$$r_1(x, z_1) \land r_2(z_1, z_2) \land \cdots \land r_K(z_{K-1}, y) \Rightarrow H(x, y), \tag{1}$$

where $r_k$ represents relations and $x$, $y$ $z_k$ represent entities in the graph. Even though this formulation is computationally efficient (see Section 3), it overlooks potential contextual information coming from local sub-graphs neighboring the entities (variables $x$, $y$, and all $z_i$). However, this kind of contextual information can be important for reasoning on knowledge graphs. Figure 1(b) shows an example. If we only know that $z$ is mother of $x$ and $y$, we are not able to infer if $y$ is a brother or sister of $x$. However, in Figure 1(c), with the contextual logical information that $\exists z'$ *is_son_of*$(y, z')$ we can infer that $y$ is a male and that $y$ should be the brother rather than the sister of $x$.

---

[1]All code and data are publicly available at `https://github.com/Glaciohound/LERP`.

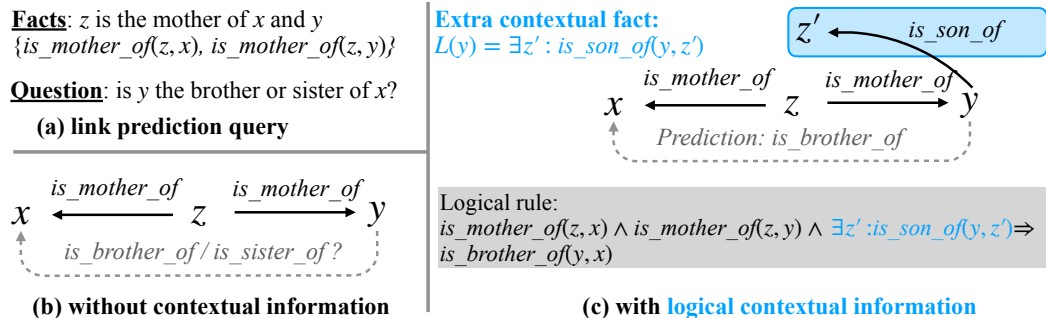

Figure 1: Incorporating contextual information of entities can benefit missing link prediction.

Recent deep neural network models based on Graph Neural Networks (GNNs) (Teru et al., 2020; Mai et al., 2021) have utilized local sub-graphs as an important inductive bias in knowledge graph completion. Although GNNs can efficiently incorporate neighboring information via message-passing mechanisms to improve prediction performance (Zhang et al., 2019; Lin et al., 2022), they are not capable of discovering explicit logical rules, and the reasoning process of GNNs is largely unexplainable.

In this paper, we propose *Logical Entity RePresentation (LERP)* to incorporate information from local sub-graphs into probabilistic logic rule learning. LERP is a logical contextual representation for entities in knowledge graph. For an entity $e$, a LERP $\boldsymbol{L}(e)$ is designed as a vector of logical functions $L_i(e)$ on $e$'s neighboring sub-graph and the enclosed relations. We then incorporate LERP in probabilistic logical rule learning methods to provide contextual information for entities. Different from other embedding learning methods, LERP encodes contextual information rather than identity information of entities. In the example discussed above in Figure 1, an ideal LERP for $y$ might contain logical functions like $L_i(y) = \exists z'\ is\_son\_of(y, z')$. Therefore, when predicting the relation $is\_brother\_of$, the rule learning model can select $L_i$ from LERP and get the rule written in Figure 1(c). In our model, LERP can be jointly optimized with probabilistic logical rules. We empirically show that our model outperforms previous logical rule learning methods on knowledge graph completion benchmarks. We also find that LERP allows our model to compete, and sometimes even exceed, strong black-box baselines. Moreover, LERP is itself an interpretable representation, so our model is able to discover more complex logical rules from data. In Section 5.4 we demonstrate that LERP can also be combinedd with embedding-learning models like TransE (Bordes et al., 2013) to construct a hybrid model that learns interpretable embeddings.

## 2 RELATED WORK

**Logical Rule Learning** This work is closely related to the problem of learning logical rules for knowledge graph reasoning, and thus also related to the inductive logic programming (ILP) field. Traditional rule learning approaches search for the logical rules with heuristic metrics such as support and confidence, and then learn a scalar weight for each rule. Representative methods include Markov Logic Networks (Richardson & Domingos, 2006; Khot et al., 2011), relational dependency networks (Neville & Jensen, 2007; Natarajan et al., 2010), rule mining algorithms (Galárraga et al., 2013; Meilicke et al., 2019), path finding and ranking approaches (Lao & Cohen, 2010; Lao et al., 2011; Chen et al., 2016), probabilistic personalized page rank (ProPPR) models (Wang et al., 2013; 2014a), ProbLog (De Raedt et al., 2007), CLP(BN) (Costa et al., 2002), SlipCover (Bellodi & Riguzzi, 2015), ProbFoil (De Raedt et al., 2015) and SafeLearner (Jain, 2019). However, most traditional methods face the problem of large searching space of logical rules, or rely on predefined heuristics to guide searching. These heuristic measures are mostly designed by humans, and may not necessarily be generalizable to different tasks.

**Differentiable Logical Rule Learning** Recently, another trend of methods propose jointly learning the logical rule form and the weights in a differentiable manner. Representative models include the embedding-based method of Yang et al. (2015), Neural LP (Yang et al., 2017), DRUM (Sadeghian et al., 2019) and RLvLR (Omran et al., 2018). Furthermore, some efforts have applied reinforcement learning to rule learning. The idea is to train agents to search for paths in the knowledge graph connecting the start and destination entities. Then, the connecting path can be ex-

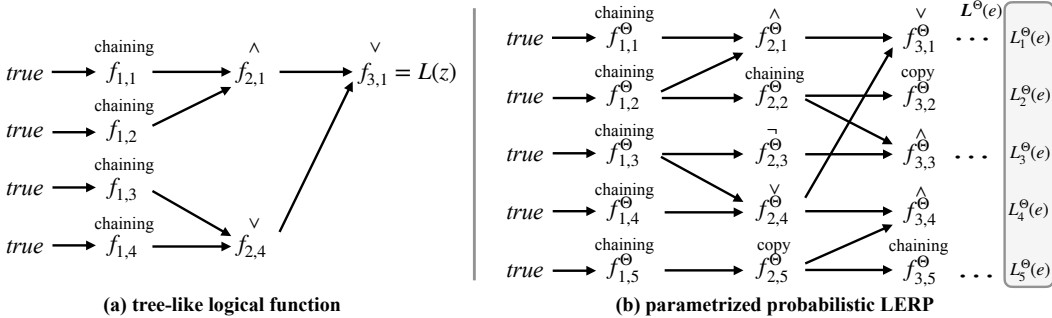

**(a) tree-like logical function**    **(b) parametrized probabilistic LERP**

Figure 2: Illustration of our construction of logical entity function and representation. (a): Each logical function is defined in a recursive manner with operations defined in Definition 1 (b): we build LERP in a feed-forward style, by constructing intermediate logical functions column by column.

tracted as a logical rule (Xiong et al., 2017; Chen et al., 2018; Das et al., 2018; Lin et al., 2018; Shen et al., 2018). These methods are mostly focused on learning chain-like Horn clauses like Equation 1. This formulation limits logical rules from using extra contextual information about local subgraphs. In contrast, we propose to use LERP to incorporate contextual information into logical rules, and are capable of modeling a more expressive family of rules.

**Embedding-Based Neural Graph Reasoning**    Our work is also related and compared to graph embedding-based methods. However, those models do not explicitly model logic rules, but reason about entities and relations over the latent space. Representative work include TransE (Bordes et al., 2013), RotatE (Sun et al., 2018), ConvE (Dettmers et al., 2018), ComplEx (Trouillon et al., 2016), TransH (Wang et al., 2014b), HolE (Nickel et al., 2016), KBGAN (Cai & Wang, 2018), TuckER (Balažević et al., 2019), Yang et al. (2015), and box embedding methods (Ren et al., 2020; Abboud et al., 2020; Onoe et al., 2021). LERP is different from these latent embeddings because LERP encodes contextual information rather than identity information of entities like in TransE. LERP is composed of vectors of interpretable logical functions, and can be applied to explicitly learning of more complex logical rules. Yang et al. (2015) also learns entity embeddings for rule learning, but the rules are still limited to chain-like rules, and the embedding is not interpretable.

## 3 PROBLEM STATEMENT

**Chain-like Horn Clauses**    are a set of first-order logic rules that have been widely studied in logical rule learning (Sadeghian et al., 2019; Yang et al., 2017). Formally, the task setting provides a knowledge graph $G = \{(s, r, o)|s, o \in \mathcal{E}, r \in \mathcal{R}\}$, where $\mathcal{E}$ is the set of entities and $\mathcal{R}$ is the set of relation types. The chain-like Horn clauses are defined as in Equation 1, where $x, y, z_k \in \mathcal{E}$ (for $k \in [1, K-1]$) and $H, r_k \in \mathcal{R}$ (for $k \in [1, K]$). The "reverse" edges $r'(z, z') = r(z', z)$ are often added to the edge type set $\mathcal{R}$. $r_1(x, z_1) \wedge \cdots \wedge r_K(z_{K-1}, y)$ is typically named the "body" of the rule.

A computational convenience of this formulation is that, if $x$ is known, the rule can be evaluated by computing a sequence of matrix multiplications. We can first order entities in $\mathcal{E}$ from 1 to $n = |\mathcal{E}|$, let $\mathbf{v}_x$ be an $n$-dimensional one-hot vector of $x$'s index, and let $\boldsymbol{A}_{r_k}$ be the adjacency matrix of $r_k$. In this way, we can compute $\mathbf{v}_x^\top \prod_{k=1}^K \boldsymbol{A}_{r_k}$ in $O(Kn^2)$ time, by multiplying in a left-to-right order from $\mathbf{v}_x^\top$ side. It is easy to verify that this result is an $n$-dimensional vector counting how many paths connect $x$ and $y$ following the relation sequence $r_1, r_2 \cdots r_K$. This formulation has been widely adopted by previous methods.

**Logical Entity Representation (LERP)**    In this work, we introduce LERP in logical rules to provide additional local contextual information for entities in the knowledge graph. A LERP for entity $e$ is formatted as a vector $\boldsymbol{L}(e) = (L_1(e), \cdots, L_m(e))$ where the dimension $m$ is a hyper-parameter. Each $L_i(e)$ is a logical function over $e$'s surrounding sub-graph. To enable efficient evaluation of $L_i(e)$, we limit $L_i(e)$ to a family of *tree-like logical functions* as defined below:

**Definition 1** *(Tree-like logical function)*

| Category | Model | WN18RR | | | | WN18 | | | |
|---|---|---|---|---|---|---|---|---|---|
| | | MRR | H@1 | H@3 | H@10 | MRR | H@1 | H@3 | H@10 |
| No Rule Learning | ComplEx | 0.44 | 0.41 | 0.46 | 0.51 | 0.941 | 0.936 | 0.936 | 0.947 |
| | TransE | 0.466 | 0.4226 | - | 0.556 | 0.495 | 0.113 | 0.888 | 0.943 |
| | ConvE | 0.43 | 0.40 | 0.44 | 0.52 | 0.943 | 0.935 | 0.946 | 0.956 |
| | LinearRE | 0.495 | 0.453 | 0.509 | 0.578 | 0.952 | 0.947 | 0.955 | 0.961 |
| | Inverse Model | 0.35 | 0.35 | 0.35 | 0.35 | **0.963** | **0.953** | 0.964 | 0.964 |
| | QuatDE | 0.489 | 0.438 | 0.509 | 0.586 | 0.95 | 0.944 | 0.954 | 0.961 |
| | MLMLM | 0.502 | 0.439 | 0.542 | 0.611 | - | - | - | - |
| | kNN-KGE | 0.579 | 0.525 | 0.604 | **0.683** | - | - | - | - |
| Rule Learning | Neural LP | 0.435 | 0.371 | 0.434 | 0.566 | 0.94 | – | - | 0.945 |
| | DRUM | 0.486 | 0.425 | 0.513 | 0.586 | 0.944 | 0.939 | 0.943 | 0.954 |
| | RNNLogic+ | 0.513 | 0.471 | 0.532 | 0.597 | - | - | - | - |
| | LERP | **0.622** | **0.593** | **0.634** | 0.682 | 0.958 | 0.932 | **0.982** | **0.987** |

Table 1: Experiment results on knowledge graph completion tasks on WN18RR and WN18.

*Given a binary predicate set $\mathcal{R}$, the family of tree-like logical functions $\mathcal{T}$ is recursively constructed with the following operations:*

1. *(True function) $f(w_0) = true$ belongs to $\mathcal{T}$.*

2. *(Chaining) $\forall\, f(w_i) \in \mathcal{T}$ and $r \in \mathcal{R}$, $f'(w_{i+1}) = \exists w_i : f(w_i) \wedge r(w_i, w_{i+1})$ belongs to $\mathcal{T}$.*

3. *(Negation) $\forall\, f(w_i) \in \mathcal{T}$, $f'(w_i) = \neg f(w_i)$ belongs to $\mathcal{T}$.*

4. *(Merging) $\forall\, f(w_i), f'(w_{i'}) \in \mathcal{T}$, let $i'' = \max(i, i') + 1$, after applying substitution $\{w_i \mapsto w_{i''}\}$ and $\{w_{i'} \mapsto w_{i''}\}$ to two functions respectively, $f''(w_{i''}) = f(w_{i''}) \wedge f'(w_{i''})$ ($\wedge$-merging) and $f''(w_{i''}) = f(w_{i''}) \vee f'(w_{i''})$ ($\vee$-merging) belong to $\mathcal{T}$.*

This definition includes arbitrarily complex functions, but in practice we limit a maximum number of operations. Intuitively, one can think of a function $f(w_i) \in \mathcal{T}$ as mimicking the shape of a local tree spanning from $w_i$ as in Figure 2(a). Specifically, this is because it gathers information together recursively through logical operations. $\boldsymbol{L}(e)$ then serves as a logical contextual representation for $e$'s local surrounding sub-graph. We then extend Equation 1 into the following formula to incorporate contextual information from $\boldsymbol{L}(e)$ ($w$ are entity variables and are interchangeable with $z$):

$$r_1(x, z_1) \wedge r_2(z_1, z_2) \wedge \cdots \wedge r_K(z_{K-1}, y) \wedge \left( \bigwedge_{z' \in \{x, y, z_1, \cdots\}} L_{i_{z'}}(z') \right) \Rightarrow H(x, y), \quad (2)$$

where $i_{z'} \in [1..m]$ is an index to select logical functions from LERP. This extension does not break the computational efficiency of chain-like Horn clauses (proofs are included in Appendix A):

**Theorem 1** *For any rule expressed in Equation 2, if $x$ is known, the evaluation of its LHS over $x$ and all $y \in \mathcal{E}$ (considering all possible $z_i$), can be determined in $O(Cn^2)$ time where $n = |\mathcal{E}|$ is the size of the entity set and $C$ is a constant related to the rule.*

## 4 FRAMEWORK

### 4.1 DERIVATION OF LERP

This section describes our computational modeling of LERP and how to incorporate it in probabilistic logic rule learning. Borrowing ideas from Yang et al. (2017); Sadeghian et al. (2019) we relax LERP to the probabilistic domain to make its results continuous. Figure 2 provides an illustration of our method. As we do not know the logical form of the functions (Figure 2(a)) beforehand, we adopt a "reduncancy" principle and calculate a matrix of intermediate logical functions $f_{j,i}^{\Theta}$ (Figure 2(b)) where $1 \leq i \leq m$ and $0 \leq j \leq T$. For example, $f_{3,1}^{\Theta}$ in 2(b) represents the same logical form as $f_{3,1}$ in 2(a). The learning process is then to discover a good hierarchy within the matrix. When $T$ is set to value larger than 1, the learned LERP can conduct multi-hop reasoning.

| | Family | | | | Kinship | | | | UMLS | | | |
|---|---|---|---|---|---|---|---|---|---|---|---|---|
| | MRR | H@1 | H@3 | H@10 | MRR | H@1 | H@3 | H@10 | MRR | H@1 | H@3 | H@10 |
| MLN | - | - | - | - | 0.351 | 0.189 | 0.408 | 0.707 | 0.688 | 0.587 | 0.755 | 0.869 |
| Boosted RDN | - | - | - | - | 0.469 | 0.395 | 0.520 | 0.567 | 0.227 | 0.147 | 0.256 | 0.376 |
| PathRank | - | - | - | - | 0.369 | 0.272 | 0.416 | 0.673 | 0.197 | 0.148 | 0.214 | 0.252 |
| MINERVA | - | - | - | - | 0.401 | 0.235 | 0.467 | 0.766 | 0.564 | 0.426 | 0.658 | 0.814 |
| NeuralLP | 0.931 | 0.880 | 0.978 | 0.994 | 0.302 | 0.167 | 0.339 | 0.596 | 0.483 | 0.332 | 0.563 | 0.775 |
| DRUM | 0.958 | 0.927 | 0.987 | 0.996 | 0.534 | 0.367 | 0.628 | 0.885 | 0.695 | 0.546 | 0.808 | 0.935 |
| RNNLogic | 0.893 | 0.862 | 0.923 | 0.928 | 0.639 | 0.495 | 0.731 | 0.924 | 0.745 | 0.630 | 0.833 | 0.924 |
| **LERP** | **0.970** | **0.947** | **0.991** | **0.997** | **0.643** | **0.499** | **0.735** | **0.931** | **0.762** | **0.646** | **0.855** | **0.942** |

Table 2: Experiment results on statistical relation learning datasets.

Specifically, we compute the tree-like probabilistic logical functions $L_i^\Theta$ parameterized by $\Theta$ by adopting a feed-forward paradigm, and compute intermediate functions $f_{j,i}^\Theta(z_{j,i})$ column by column. The first column contains only $true$ functions $f_{0,i}^\Theta(z_{0,i}) = true$. For $j > 0$, each $f_{j,i}^\Theta(z_{j,i})$ is constructed from the previous column by 6 possible operations: $true$ $function$, $chaining$, $negation$, $\wedge$-$merging$, and $\vee$-$merging$ from Definition 1, as well as an additional $copy$ operation that simply copies $f_{j-1,i}^\Theta(z_{j-1,i})$. Note that the result of $f_{j,i}^\Theta(z_{j,i})$ over $\mathcal{E}$ can be represented as a $|\mathcal{E}|$-dimensional vector $\mathbf{v}_{j,i} \in [0,1]^{|\mathcal{E}|}$. Taking this notation, the formulae for the 6 possible types of $f_{j,i}^\Theta(z_{j,i})$ are:

$$\mathbf{v}_{j,i;\text{true}} = \mathbf{1} \qquad\qquad \mathbf{v}_{j,i;\text{chaining}} = \text{clamp}\left(\mathbf{v}_{j-1,i}^\top \sum_{r\in\mathcal{R}} \alpha_{r,j,i}\mathbf{A}_r\right)^\top$$

$$\mathbf{v}_{j,i;\neg} = \mathbf{1} - \mathbf{v}_{j-1,i} \qquad\qquad \mathbf{v}_{j,i;\text{copy}} = \mathbf{v}_{j-1,i}$$

$$\mathbf{v}_{j,i;\wedge} = \mathbf{v}_{j-1,i} \odot \sum_{i'=1}^{m} \beta_{j,i,i'}^{\wedge}\mathbf{v}_{j-1,i'} \qquad\qquad \mathbf{v}_{j,i;\vee} = \mathbf{1} - (\mathbf{1} - \mathbf{v}_{j-1,i}) \odot (\mathbf{1} - \sum_{i'=1}^{m} \beta_{j,i,i'}^{\vee}\mathbf{v}_{j-1,i'})$$

where $\odot$ denotes Hadamard product, and $\text{clamp}(x) = 1 - e^{-x}$ clamps any positive number to range $[0,1]$. Note that we constrain that $\alpha_{r,j,i} > 0$, $\sum_r \alpha_{r,j,i} = 1$, $\beta_{j,i,i'}^* > 0$, $\sum_{i'} \beta_{j,i,i'}^* = 1$. These 6 types of scores are then gathered by a probability distribution $p_{j,i,*}$:

$$\mathbf{v}_{j,i} = p_{j,i,1}\mathbf{v}_{j,i;\text{true}} + p_{j,i,2}\mathbf{v}_{j,i;\text{chaining}} + p_{j,i,3}\mathbf{v}_{j,i;\neg} + p_{j,i,4}\mathbf{v}_{j,i;\text{copy}} + p_{j,i,5}\mathbf{v}_{j,i;\wedge} + p_{j,i,6}\mathbf{v}_{j,i;\vee}$$

Note that, specially in the second column, we only allow chaining $f_{1,i}^\Theta$, because only chaining produces meaningful logical functions out of the first column of $true$ functions. Thereby $\mathbf{v}_{j,i}$ represents the result of the probabilistic logical function $f_{j,i}^\Theta$ on all entities. $p, \alpha, \beta$ are all obtained by a softmax operation over some learnable parameters $p', \alpha', \beta'$. These parameters belong to $\Theta$. There are a total of $T + 1$ columns of intermediate functions. Finally we use the last column of functions $f_{T,i}^\Theta$ as $L_i^\Theta$, and $\boldsymbol{L}^\Theta(e) = (L_1^\Theta(e), \cdots, L_m^\Theta(e))$ is our logical entity representation LERP on $e$.

Next we explain the computation of chain-like probabilistic logical rules, and how to further incorporate LERP. In Yang et al. (2017), they model a probabilistic selection of logical predicates, so that the rules can be differentiably evaluated. Suppose for the chain-like rules in Equation 1, the predicate of $r_k$ is selected according to a distribution $a_{r,k}$. Then the probability that the clause body is true between each $x, y \in \mathcal{E}$ can be given by

$$\prod_{k=1}^{K} \left(\sum_{r\in\mathcal{R}} a_{r,k}\mathbf{A}_r\right) \tag{3}$$

Now we want to also add a logical function constraint $L_{i_k}^\Theta$ from $\boldsymbol{L}^\Theta$. Denote $\boldsymbol{L}^\Theta(\mathcal{E})$ as an $m \times |\mathcal{E}|$ matrix by iterating $\boldsymbol{L}^\Theta(e)$ over all $e \in \mathcal{E}$. We can select $L_{i_k}^\Theta$ from $\boldsymbol{L}^\Theta$ according to a probabilistic distribution $\boldsymbol{\rho_k}$ over $[1..m]$. Then the results of $L_{i_k}^\Theta(z_k)$ on $\mathcal{E}$ can be written in vector form $\boldsymbol{L}^\Theta(\mathcal{E})^\top \boldsymbol{\rho_k}$. Finally, the evaluation of the body in Equation 2 can be written as

$$\prod_{k=1}^{K} \left(\sum_{r\in\mathcal{R}} a_{r,k}\mathbf{A}_r\right) \text{diag}\left(\boldsymbol{L}^{\Theta\top}\boldsymbol{\rho_k}\right) \tag{4}$$

| weight | logical rule |
|---|---|
| 0.10 | $(\nexists z_2 : sister(z_2, x)) \wedge mother(z_1, x) \wedge son(y, z_1) \Rightarrow brother(y, x)$ |
| 0.016 | $mother(z_1, x) \wedge mother(z_1, y) \wedge (\nexists z_3 : sister(y, z_3)) \Rightarrow brother(y, x)$ |
| 0.11 | $son(z_1, x) \wedge brother(y, z_1) \Rightarrow son(y, x)$ |
| 0.004 | $(\nexists z_2 : sister(z_2, x)) \wedge brother(z_1, x) \wedge (\nexists z_3 : sister(z_3, z_1)) \wedge mother(y, z_1) \Rightarrow son(x, y)$ |
| 0.039 | $wife(x, y) \Rightarrow husband(y, x)$ |
| 0.024 | $mother(x, z_1) \wedge father(y, z_1) \Rightarrow husband(y, x)$ |

Table 3: A set of logical rules discovered from the Family dataset.

| weight | logical function | weight | logical function |
|---|---|---|---|
| 0.99 | $\nexists z_1 : sister(e, z_1)$ | 0.56 | $\exists z_1 : brother(e, z_1) \wedge \exists z_2 : aunt(z_2, e)$ |
| 0.99 | $\nexists z_1 : brother(e, z_1)$ | 0.46 | $\exists z_1 : aunt(e, z_1) \wedge \exists z_2 : daughter(z_2, e)$ |
| 0.96 | $\nexists z_1 : sister(z_1, e)$ | 0.39 | $\nexists z_1 : uncle(e, z_1)$ |
| 0.68 | $\exists z_1 : sister(e, z_1) \wedge \exists z_2 : sister(z_2, e)$ | 0.25 | $\exists z_1 : brother(e, z_1) \wedge \exists z_2 : daughter(z_2, e)$ |

Table 4: The list of logical functions $L_i(e)$ learned in LERP $\boldsymbol{L}(e)$.

This is the evaluation for a single logical rule. In practice we can have multiple rules to predict one target relation (e.g., in DRUM, there are up to 4), and there are multiple types of target relations in the dataset. So, we learn individual sets of parameters $a_{r,k}$ and $\boldsymbol{\rho}_k$ for different rules, but the parameter $\Theta$ for LERP is shared across rules.

## 4.2 REASONING AND TRAINING

In the task of knowledge graph completion, the model is given a set of queries $r(h, ?)$ and required to find the entity $t \in \mathcal{E}$ that satisfies relation $r(h, t)$. During reasoning and training, we denote $\mathbf{h}$ as the one-hot vector of entity $h$, and multiply it with Equation 4 to calculate

$$\hat{\mathbf{t}}^\top = \mathbf{h}^\top \prod_{k=1}^{K} \left( \sum_{r \in \mathcal{R}} a_{r,k} \boldsymbol{A}_r \right) \text{diag} \left( \boldsymbol{L}^{\Theta^\top} \boldsymbol{\rho}_k \right)$$

This then can be calculated as a sequence of vector-matrix multiplications. The resulting vector is indicating which entity $t$ is satisfying the query $r(h, ?)$. For training criterion, we use the cross entropy between the normalized prediction $\frac{\hat{\mathbf{t}}}{\|\hat{\mathbf{t}}\|_1}$ and ground-truth one-hot vector $\mathbf{t}$. The loss is differentiable with respect to all the parameters in our model, so we can apply gradient-based optimization methods such as Adam (Kingma & Ba, 2015). During evaluation, we use $\hat{\mathbf{t}}$ as scores and sort it to obtain a ranked entity list for calculating mean reciprocal rank (MRR), Hits, etc.

## 5 EXPERIMENTS

### 5.1 KNOWLEDGE GRAPH COMPLETION

**Datasets** We follow previous works (Yang et al., 2017; Sadeghian et al., 2019) and evaluate on the Unified Medical Language System (UMLS), Kinship, and Family datasets (Kok & Domingos, 2007) as well as the WN18 (Bordes et al., 2013) and WN18RR (Dettmers et al., 2018) datasets. The task is to predict missing edges in theses dataset, and is named "knowledge graph completion" on knowledge graphs like WN18RR and WN18 and "statistical relation learning" on smaller datasets like UMLS, Kinship and Family. For all experiments, we evaluate under the setting of using no external data. For Kinship and UMLS, we follow the data split from Qu et al. (2020), and for Family, we follow the split used by Sadeghian et al. (2019).

For rule learning baselines, we compare with Markov Logic Networks (MLN) (Richardson & Domingos, 2006), Boosted RDN (Natarajan et al., 2010), and path ranking (PathRank) (Lao & Cohen, 2010) as well as other differentiable rule learning baselines such as NeuralLP (Yang et al., 2017) and DRUM (Sadeghian et al., 2019). We also include a reinforcement learning method MINERVA (Das et al., 2018) and an expectation-maximization method RNNLogic (Qu et al., 2020) in

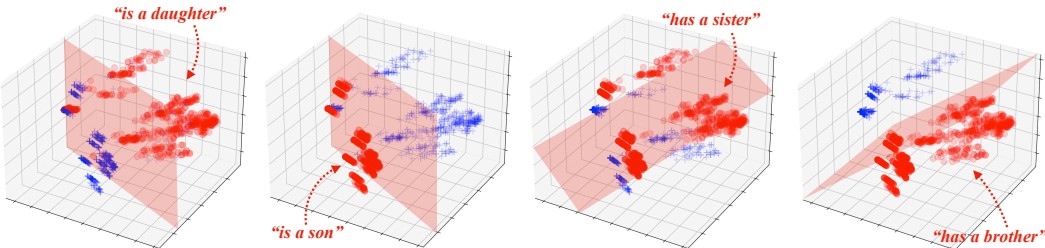

Figure 3: Visualization of the space of LERP on the Family dataset. In each figure, we color as red entities of different relative roles (e.g., people who *are daughters*, *sons*, or *have sisters* or *brothers*).

baselines. On knowledge graph completion task, we further compare with the following powerful black-box embedding-based models: ComplEx (Trouillon et al., 2016), TransE (Bordes et al., 2013), ConvE and Linear Model (Dettmers et al., 2018), LinearRE (Peng & Zhang, 2020), QuatDE (Gao et al., 2021), MLMLM (Clouatre et al., 2021), and kNN-KGE (Zhang et al., 2022).

For evaluation, following the setting in Sadeghian et al. (2019), we we adopt filtered ranking and evaluate models on mean reciprocal rank (MRR) as well as the Hits@{1, 3, 10} metrics. We search for hyper-parameters according to validation set performance, and finally adopt the hyperparameters dimension $m = 80$ and depth $T = 2$ for LERP, and the maximum rule length of $K = 3$. We use 4 rules for WN18RR and WN18 and Family and 200 rules per target relation type for Kinship and UMLS dataset. Our model is trained for 4 epochs for WN18RR and WN18 and 10 epochs for Family, Kinship, and UMLS. As DRUM runs on fewer number of rules by default, we re-run DRUM with larger number of rules to match the configuration of our model and RNNLogic. For optimization, we use Adam with a learning rate of 0.1, $\beta_1 = 0.9$, and $\beta_2 = 0.999$.

The results for are demonstrated in Tables 1 and 2. Compared with other rule learning methods, our model obtains higher performance across the board. This validates the importance of contextual information provided by LERP, especially after considering that this is the only difference in model design between DRUM and our model. Black box models, which do not conduct rule learning, generally achieve higher scores than rule learning methods. This is both because of the strong fitting power of black-box models and because rule learning models also need to explicitly learn logical rules. However, our model achieves performance comparable or even superior to black-box baselines. Even on metrics where LERP's performance falls short of that of state-of-the-art models (like Hits@10 on WN18RR dataset), the difference is marginal. We attribute this to the property of LERP modeling a more expressive set of logical rules, allowing it to achieve data-fitting power similar to black-box neural models.

## 5.2 INTERPRETABILITY OF LERP AND LOGICAL RULES

An important advantage of logical rules is that they help humans to better comprehend the reasoning process of the algorithm. In this subsection, we provide a case study on the interpretability of LERP and our model on Family dataset.

In Table 3, we sort the logical rules according to their weights, and show the top-ranked logical rules for three relations. We can see that these logical rules are more complex than chain-like Horn clauses as in Equation 1. For example, in the second row, the logical function $\nexists sister(y, z_3)$ provides contextual information that $y$ is not sister of anybody. [2] Therefore, $y$ is more likely to be the brother than sister of $x$. Similarly, in the fourth row, $\nexists z_3 : sister(z_3, z_1)$ provides information that $z_1$ has no sister, so $x$ can only be the brother of $z_1$ and thus the son of $y$. We can also observe some verbosity in the learned rules, like the logical function $\nexists z_2 : sister(z_2, x)$ in the first row. This is not necessary since $son(y, z_1)$ is enough for predicting that $y$ is the brother of $x$. But this kind of verbosity is inevitable since rules of the same meaning are equivalent from the model's perspective.

Since LERP is itself an interpretable representation, we can inspect it directly in Table 4. We sort the logical functions $L_i^\Theta$ according to the weights, and filter out the repeated functions. Some logical functions are related to the gender information of entities. For example, $\nexists sister(e, z_1)$ might suggest

---

[2]Note that in the Family dataset ontology, {relation}$(a, b)$ means that $a$ is the {relation} of $b$ (e.g., $son(a, b)$ means $a$ is $b$'s son).

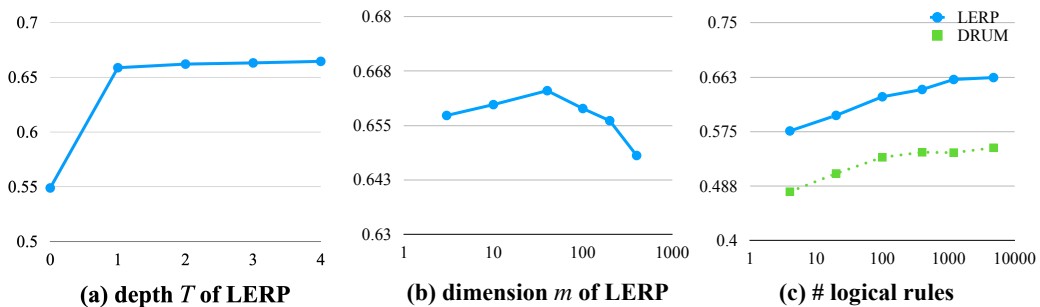

Figure 4: Analysing LERP's performance under different configurations.

(although with uncertainty) that $e$ is a male. Other logical functions are longer and contain more complex information of entities. For instance, $\exists z_1 : brother(e, z_1) \wedge \exists z_2 : daughter(z_2, e)$ tells us $e$ is a brother and has a daughter, which might help predict the nephew/aunt-uncle relationship between his sibling and daughter.

We notice that the weights of some interpreted logical functions are not close to 1.0, and the actual utility of some logic functions are not easy to guess merely from the interpreted logic form. To better understand the space of LERP representation, we conduct 3-dimensional principle component analysis (PCA) on LERP vectors of entities in the graph. We select entities with some relational roles such as *being a daughter*, *being a son*, *having a sister*, or *having a brother*, and mark them with red circles in contrast to the blue pluses of the other entities. For better visualization, we also plot the plane dividing them from the rest of entities. The results are shown in Figure 3. We find that the lower-dimensional space of LERP is well organized according to these roles. Note that the role of *having a brother* is not explicitly stated in the learned logical functions in Table 4. We hypothesize that it might come from a combination of other logical functions like $\exists z_1 : brother(e, z_1) \wedge \exists z_2 : aunt(z_2, e)$ and $\exists z_1 : brother(e, z_1) \wedge \exists z_2 : daughter(z_2, e)$. This demonstrates that logical functions in LERP can be combined to represent extra logical information.

## 5.3 MODEL ANALYSIS

**Is contextual information from LERP useful?** This work is motivated by including contextual information for rule learning. To validate our hypothesis, we vary the depth of LERP from 0 to 4 and observe the performance. This hyper-parameter controls the complexity of logical functions learned in LERP. When depth is 0, our model does not use contextual information from LERP and defaults back to the DRUM baseline. Results are shown in Figure 4(a). We see that even with a depth of 1, LERP boosts our model's accuracy by a large margin compared with the depth of 0. Similar conclusion can also come from the comparison between DRUM and LERP in Table 2 and 1, because DRUM can be viewed as an ablation study counterpart of LERP by removing contextual information from LERP.

**Is a higher dimension of LERP always better?** The answer is no. Similar to embedding-based models like TransE, a larger dimension may cause LERP to overfit the training data and perform worse on the test set. In Figure 4(b), we set the width of LERP to $\{3, 10, 40, 100, 200, 400\}$. We observe a $\cap$-shaped curve with 40 giving optimal test performance.

**Does the model benefit from larger number of logical rules?** The default number of logical rules can be different across models. For example, RNNLogic uses 100∼200 rules per relations, but DRUM only uses 1∼4 rules by default. To better analyze the model's performance, we vary the number of rules in our model. We also compare with DRUM. In Figure 4(c), both models' performance increase with larger number of rules, but with diminishing returns (note the logarithmic scale of the x-axis). Our model achieves higher scores even with fewer number of rules. We attribute this to LERP helping to learn higher quality rules.

## 5.4 AUGMENTING EMBEDDING LEARNING MODELS WITH LERP

LERP is designed to learn an interpretable logical representation for entities in graphs. One question then arises: can LERP work together with other embedding-learning methods such as

| logical function | relations | logical function | relations |
|---|---|---|---|
| $\exists z : niece(e, z)$ | daughter,niece | $\exists z : aunt(e, z) \vee \exists z : niece(e, z)$ | mother |
| $\nexists z : niece(e, z)$ | father,husband,son | $\exists z : aunt(e, z) \wedge \exists z : niece(e, z)$ | |
| $\nexists z : uncle(e, z)$ | aunt,sister | $\exists z_1, z_2 : mother(z_1, e) \wedge : niece(z_1, z_2)$ | brother,nephew |

Table 5: The list of logical functions $L_i(e)$ learned when optimizing TransE+LERP. We project the learned translation vectors $l$ back to the LERP space and identify most correlated logical functions.

TransE (Bordes et al., 2013), to combine the logic structure in LERP and strong fitting power of TransE? Inspired by this idea, we design a hybrid model TransE+LERP. We use the training framework and training criterion on TransE, but the entities' embeddings are now defined as $Embed'(e) = Embed(e) + R\boldsymbol{L^{\Theta}}(e)$. $Embed(e)$ is set as a 20-dimensional entity-specific embedding. We also set the width and depth of LERP to be 8 and 2. $R$ is a $20 \times 8$ matrix to transform LERP to the embedding space. Besides the loss in original TransE framework, we add a regularization term $\eta \frac{\sum_e \|Embed(e)\|_2}{\sum_e \|R\boldsymbol{L^{\Theta}}(e)\|_2}$ to encourage LERP to encode important information during optimization. We also introduce an advanced version TransE+LERP+entropy by adding an extra entropy regularization term $\gamma H(\Theta)$ on LERP's weights. This term is added to loss and encourage LERP to assign more weights to fewer logical rules. In experiments, we set $\eta = 10$ and $\gamma = 1$.

We evaluate the models on Family dataset and results are presented in Figure 5. The vanilla TransE model tends to overfit the training set ($>95\%$ training set accuracy) because Family dataset contains a limited number of training triplets. TransE+LERP achieves higher test set performance. We hypothesize that LERP provides logical regularization on the embedding $Embed'(e)$ and makes it more generalizable. After adding the entropy regularization term, the performance further improves. We attribute this to that the regulation encourages learning more explicit logical functions, which serves as a prior in embedding learning.

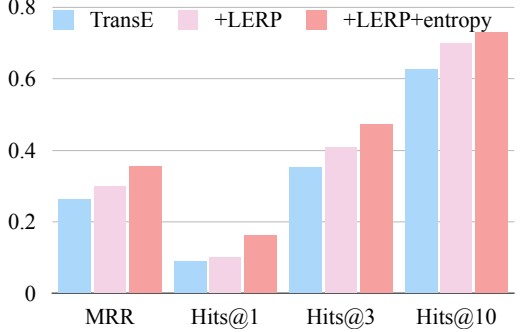

Figure 5: Incorporating LERP representation improves TransE's performance on Family dataset.

We also study the interpretability of the hybrid model. In Table 5 we list the logical functions that are learned by TransE+LERP. There are 2 repetitions in logical functions which are removed. We also manage to identify the correlation between relations in TransE and logical functions. In TransE, each relationship has a vector $l$ to denote a translation in the embedding space. $l$ resides in the embedding space, and we want to find a vector $l'$ in LERP space so that $Rl' \approx l$. As $R$ is not square matrix, we use $R$'s Moore–Penrose inverse $R^+$ to project the translation vectors to $R^+l$. We then look for dimensions with highest values in $R^+l$, and results are listed in the 2nd and 4nd columns in Table 5. We can observe that these functions encode rich information. For example, relations like *daughter* and niece are associated with $\exists z : niece(e, z)$ which suggests a female gender and lower rank in the family tree.

## 6 Conclusions and Future Work

In this work, we propose a logical entity representation (LERP) and demonstrate that we can incorporate contextual information of local subgraphs to probabilistic logical rule learning. Therefore we are able to learn a more expressive family of logical rules in a fully differentiable manner. In experiments on graph completion benchmarks, our model outperforms rule learning baselines and competes with or even exceeds state-of-the-art black box models. LERP is itself an interpretable representation of entities. We also find that LERP can be combined with latent embedding learning methods to learn interpretable embeddings. For future work, we expect that LERP can be combined with a wider range of neural networks to provide logical constraints and learn interpretable embeddings.

REPRODUCIBILITY STATEMENT

The datasets, split and the configuration of the model are introduced in Section 5.1. The statistics of the datasets used are also included in Appendix D. We also conduct analysis of the model behavior in Section 5.3. In the additional experiments of combining TransE and LERP, we provide details of implementation in Section 5.4, including the hyper-parameters and rationale for some technical decisions. For Theorem 1, we provide proof in Appendix A.

ACKNOWLEDGMENTS

We would like to thank anonymous reviewers for valuable comments and suggestions. This work was supported in part by US DARPA KAIROS Program No. FA8750-19-2-1004 and AIDA Program No. FA8750-18-2-0014. The views and conclusions contained in this document are those of the authors and should not be interpreted as representing the official policies, either expressed or implied, of the U.S. Government. The U.S. Government is authorized to reproduce and distribute reprints for Government purposes notwithstanding any copyright notation here on.

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

## A  PROOF OF THEOREM 1

More specifically, we will prove that when $C = K(1 + D)$, where $K$ is the length of the rule in Equation 2 and $D$ is the maximum number of used operations in Definition 1 for constructing each $L_{i_{z'}}$, the computation time for evaluating $x$ and any $y \in \mathcal{E}$ can be bounded by $O(Cn^2)$ where $n = |\mathcal{E}|$. This is an asymptotically tight bound, because reading all the edge information naturally requires a lower bound of $\Omega(\min(C, |\mathcal{R}|)n^2))$ time.

First it is easy to see that all tree-like functions only have one free variable. We will bound the computation time of evaluating tree-like functions $f(e) \in \mathcal{T}$ over all entities in $\mathcal{E}$. Let $\mathbf{v}(f)$ be the vector of scores of $f$ on $\mathcal{E}$, and $d(f)$ be the number of operations used for constructing $f$, we recursively prove that $\mathbf{v}(f)$ can be calculated in $O(d(f)n^2)$ time. First for an all true function $d(f) = 1$, $\mathbf{v}(f)$ contains only 1 can be directly written in $O(n)$ time. For a negation operator $f = \neg f'$, $\mathbf{v}(f)$ only needs negating $\mathbf{v}(f')$ and takes $O((d(f) - 1)n^2) + n \leq O(d(f)n^2)$ time. Similarly for a merging operation $f = f_1 \wedge f_2$, $\mathbf{v}(f)$ can be derived by processing elements of $\mathbf{v}(f_1)$ and $\mathbf{v}(f_2)$ sequentially and needs $O((d(f) - 1)n^2) + 2n \leq O(d(f)n^2)$ time. Finally for a $f$ constructed by chaining $f'$ and relation $r$, $\mathbf{v}(f)$ is just $\mathbf{v}(f')^\top \mathbf{A}_r$ in binary field and takes $O((d(f) - 1)n^2) + n^2 \leq O(d(f)n^2)$ time.

So getting result vectors for all tree-like logical functions takes no more than $O(KDn^2)$ time. We then plus the time for evaluating the main body in Equation 2. we start from a one-hot vector $\mathbf{v} = \mathbf{v}_x$. After chaining each relation $r_k$, the partial evaluation of the logical rule $r_1(x, z_1) \wedge r_2(z_1, z_2) \wedge \cdots \wedge r_k(z_{k-1}, y) \wedge \left( \bigwedge_{z' \in \{x, z_1, \cdots, z_{k-1}\}} L_{i_{z'}}(z') \right)$ can be computed by $\mathbf{v}\top \leftarrow \mathbf{v}\top \mathbf{A}_{r_k}$. Applying the new constraint $L_{i_{z_k}}(z_k)$ is another element-wise logical and between $\mathbf{v}$ and $\mathbf{v}(L_{i_{z_k}})$ which takes $O(n)$ time. Summing together, the total time for evaluating the whole rule can be bounded by $O(KDn^2 + Kn + Kn^2) \leq O(K(1 + D)n^2)$.

## B  EVENT PREDICTION

| Dataset | Model | MRR | HITS@1 |
|---------|-------|-----|--------|
| **General** | Event Language Model (Rudinger et al., 2015) | 0.367 | 0.497 |
| | Sequential Pattern Mining (Pei et al., 2001) | 0.330 | 0.478 |
| | Human Schema  (Li et al., 2021) | 0.173 | 0.205 |
| | Event Graph Model (Li et al., 2021) | **0.401** | 0.520 |
| | LERP | 0.347 | **.672** |
| **IED** | Event Language Model (Rudinger et al., 2015) | 0.169 | 0.513 |
| | Sequential Pattern Mining (Pei et al., 2001) | 0.138 | 0.378 |
| | Human Schema  (Li et al., 2021) | 0.072 | 0.222 |
| | Event Graph Model (Li et al., 2021) | 0.223 | 0.691 |
| | LERP | **0.242** | **0.978** |

Table 6: Experiment results on ending event prediction.

| Dataset | Split | #doc | #graph | #event | #arg | #rel |
|---------|-------|------|--------|--------|------|------|
| **General** | **Train** | 451 | 451 | 6,040 | 10,720 | 6,858 |
| | **Dev** | 83 | 83 | 1,044 | 1,762 | 1,112 |
| | **Test** | 83 | 83 | 1,211 | 2,112 | 1,363 |
| **IED** | **Train** | 5,247 | 343 | 41,672 | 136,894 | 122,846 |
| | **Dev** | 575 | 42 | 4,661 | 15,404 | 13,320 |
| | **Test** | 577 | 45 | 5,089 | 16,721 | 14,054 |

Table 7: Dataset statistics for event prediction datasets  (Li et al., 2021) .

One driving force for humans to understand logic is to interpret the past and predict the future. Going beyond instincts, logic provides humans with the ability of explainable and reasoning-based prediction. In this section, we evaluate LERP's ability on an ending-event prediction task. This task is different from the link completion task of Subsection 5.1. The goal is to deduce the existence of

future events given an input event graph, typically produced by human annotation or information extraction from news or stories.

**Benchmark Datasets**   We conduct experiments on the graph-based event prediction datasets from Li et al. (2021), including a general scenario (General Schema Learning Corpus) and a more specific bombing scenario (IED Schema Learning Corpus). More specifically, the General Schema Learning Corpus is released by Linguistic Data Consortium (LDC2020E25) – including 82 types of events, such as Disease Outbreak and Shop Online. The original improvised explosive device (IED) dataset was collected by Li et al. (2021). They conduct a two-round annotation process with linguists to ensure quality of the annotated schemas/graphs. The dataset statistics can be found in Table 7.

**Evaluation Metrics and Training Objective**   Following prior work (Li et al., 2021), we adopt MRR (Mean Reciprocal Rank) and HITS@1 as evaluation metrics. An event is predicted correctly if the event type matches one of the ending event types in the gold-standard event graph. Considering that there can be multiple ending events in one instance graph, we rank event types with their prediction scores. The model adopts multiple output heads, and uses categorical cross entropy for training objective.

**Baselines**   Event Language Model (Rudinger et al., 2015) learns the probability of temporal event sequences, and obtain the event sequences prediction from language model. Sequential Pattern Mining (Pei et al., 2001) is a classic method that explores prefix-projection in sequential pattern mining. The most frequent mined patterns/event graphs are used for predicting the ending event. Event Graph Model (Li et al., 2021) is the state-of-the-art model on the ending event prediction task. It proposes a temporal event graph model that predicts the future event type given the current (incomplete) event graph. Finally, we also add human schema as a reference (Li et al., 2021). We match expert-created event schemas to test data graphs and fill in the matched ending event type.

**Results and Analysis**   In Table 6, we present the results on these two datasets. We find that all of the learning-based methods outperform pattern mining-based method (Pei et al., 2001) on both datasets. Our model further outperforms the baselines on most evaluation categories including the Event Graph Model on HITS@1. We attribute the inferior performance of the human schema matching baseline to that real event graphs are highly variational, so fixed schemas do not generalize well to the test graphs.

## C   LOGICAL EVENT EQUATION INDUCTION

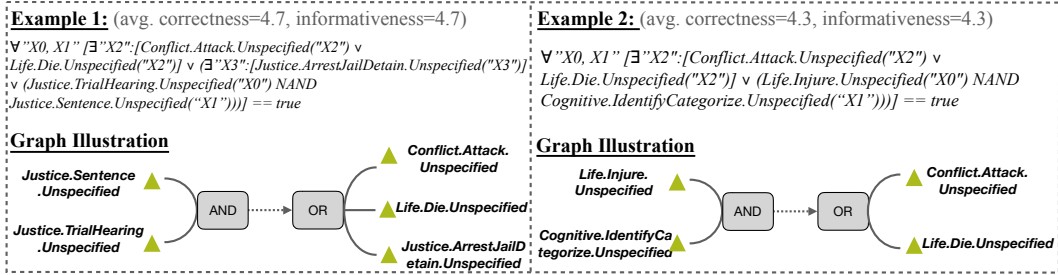

Figure 6: Induced event relation examples from IED (Li et al., 2021).

One important feature of LERP is its ability to learn explainable first-order logical formulae. In this section, we train the model to induce logical relations among events. Specifically, we use IED dataset (Li et al., 2021) for training. Specifically, we adopted contrastive training to encourage LERP to distinguish positive (real) and negative (incomplete) event graphs. The training dataset is used directly as the positive graph. We randomly sample an event graph instance, and randomly remove 30 event nodes to build a negative graph. The negative graph is then upsampled to match the number of all positive graphs. The training objective of LERP is the contrastive loss between positive and negative graphs.

|            | Family | UMLS | Kinship |
|------------|--------|------|---------|
| **#Triplets**  | 28356 | 5960 | 9587 |
| **#Relations** | 12    | 46   | 25   |
| **#Entities**  | 3007  | 135  | 104  |

|         | #Relation | #Entity | #Train | #Valid | #Test |
|---------|-----------|---------|--------|--------|-------|
| **WN18**   | 18 | 40,943 | 141,442 | 5,000 | 5,000 |
| **WN18RR** | 11 | 40,943 | 86,835  | 3,034 | 3,134 |

Table 8: Dataset statistics for statistical relation learning (Sadeghian et al., 2019) tasks.

Table 9: Dataset statistics for knowledge graphs. WN18RR is a more challenging version of WN18 by filtering out some data bias.

Induced examples are shown in Figure 6. We rank and filter the induced logical equations by the model confidence, and let human annotators to assess the quality of induced equations. As model are able to induce logical formulae in arbitrary form, we provide a visual illustration for each logic formula, so as to unify the form and provide easier human interpretation. These induced equations demonstrate LERP's ability of learning complex relations from data, which are highlt scored by humans.

## D  DATASET STATISTICS

We also list the dataset statistics in Table 9. Family, UMLS and Kinship are smaller scale datasets. WN18 and WN18RR contains a larger number of entities and relations.

## E  DISCUSSION OF COMPARISON WITH NLIL

A closely relatd work to LERP is NILI (Yang & Song, 2019), which also claims to learn tree-like logical functions. A major difference between LERP and NLIL is that, LERP is able to model more complex graph structure in logical rules, and encode more contextual information about nodes in the middle of chains connecting head and tail nodes. This is also the main motivation in our work. Although NLIL also aims at going beyond chain-like rules, its definition of "tree-like rules" is indeed very different from ours (which is also mentioned in your comments). More specifically, the rules in NLIL are formulated as multiple chain-like rules combined together logically. As we will explain in the following sub-points, this formulation still overlooks contextual information for middle nodes in logical chains, and reduce to simple baselines like NeuralLP(Yang et al., 2017) in some settings. This problem can be viewed from the following perspectives:

**Formulation of rules**  NLIL has three parts working hierarchically. (i) The first part is responsible for searching among unary predicates $\mathcal{U}$ and binary predicates $\mathcal{B}$. (ii) The second part searches for "primitive statements", which are essentially chain-like rules with either one free variable $x$ at an end or two free variables $x$ and $x'$ at two ends. This can be seen from the definition of $\psi_k$ (Equation(9) in NLIL paper): $\psi_k(x, x')$ can be written in one of the two cases:

$$\sigma(\mathbf{1}^\top \mathbf{M}_k^\top \prod_{t'=1}^{T} \mathbf{M}^{(t')} \mathbf{v}_{\mathbf{x}'}),$$

or

$$\sigma(\mathbf{v}_{\mathbf{x}}^\top \prod_{t=1}^{T} {\mathbf{M}^{(t)}}^\top \mathbf{M}_k^\top \prod_{t'=1}^{T} \mathbf{M}^{(t')} \mathbf{v}_{\mathbf{x}'}).$$

Both cases belong to the family of chain-like rules described in Equation 1 and 3 in our paper. (1.a.iii) The final step logically combines primitive statements, which applies logical operators $\{\wedge \vee \neg\}$ over chain-like rules. This steps introduces logical complexity, but does not change the chain form in the graph to encode more complex graph struture information. That is, the instantiation of the rules in the graph are still multiple chains spanning from $x$ or connecting $x$ and $x'$. Similarly, chain-like rule baselines DRUM (Sadeghian et al., 2019) and NeuralLP (Yang et al., 2017) also models disjunction of multiple chains. Our formulation, however, is able to model more complex graph structures (trees or branched chains). As shown from the following two empirical observations, NLIL's formulation can limit the expressiveness of its rule learning.

**Example rules**    NLIL presents examples of learned logical rules in its Table 4. Its rules can still be parsed as logical combination of multiple chains. For example, the first rule for $\text{Person}(X)$ can be seen as logical disjunction $\vee$ combining these chains: $\text{chain}_1 : \text{Shirt}(Y_1) \wedge \text{Wearing}(Y_1, X)$, $\text{chain}_2 : \text{Pants}(Y_2) \wedge \text{Wearing}(Y_2, X)$ and $\text{chain}_3 : \text{Street}(Y_3) \wedge \text{Wearing}(Y_3, X)$. The same also applies to other examples in Table 4 in NLIL. Baselines DRUM and NeuralLP also study learning multiple chain-like logical rules with final scores summed together, which models an implicit conjunction operation $\vee$ of these rules.  For example, one can convert DRUM Table 7 examples equivalently to a conjunction form: $\text{wife}(A, B) \leftarrow \text{husband}(B, A) \vee (\text{mother(A, C), father(B, C)}) \vee (\text{son}(C, A) \wedge \text{father}(B, C))$. NLIL's logical formulation is only different from NeuralLP in that NLIL allows for $\neg, \wedge$ to combine chains.

**Experiment results**    as argued in the above two points, NLIL is similar to chain-like rules in modelling graph structures.  Therefore, in knowledge graph completion experiments (Table 1 in NLIL paper), NLIL's absolute scores are still similar (or inferior) to chain-like rule learning baselines like NeuralLP, with gains no more than 0.01 absolute value. This also partially validates the argument that NLIL's expressiveness in graph reasoning domain is still similar to chain-like rules learning.

## F    SPEED OF RULE LEARNING

Traditional ILP methods such as FOIL (Quinlan & Cameron-Jones, 1995) is able to learn thousands to millions of logic rules using no more a few seconds per rule.  A question then arises: how fast can differentiable rule learning approaches learn logic rules?  Normally, differentiable rule learning methods for knowledge graph completion (including our approach as well as RNNLogic and DRUM) will not set the number of logical rules to very high. A typical number usually ranges from 3 or 4 to a few hundreds of rules. Main reason is that these work focus more on downstream evaluation metrics, such as knowledge graph completion accuracy. Moreover, these metrics improve only marginally when the number of rules increases to very high (see Figure 4(c) in our submission). Another consideration is that, a larger number of logic rules increases the computation overhead, because these rules are executed in parallel in experiments. Our 16GB CUDA memory space maximally allows for 3200 rules to be run together on Kinship and UMLS dataset, but doubles the running time compared with 200 rules.  So in experiments, we just choose a number (200 in this paper for Kinship and UMLS) that work reasonably well (selected by validation performance) and balances running speed, while also being comparable to previous work.

More specifically, on Kinship dataset for example, our model trains for 2.8 minutes and gets 5200 rules (200 rules / target predicate $\times$ 26 target queries).  In comparison, DRUM (Sadeghian et al., 2019) gets the same number of rules in 1.2 minutes, while RNNLogic uses 10 minutes.  They also tend to get duplicated rules for the same reason.

## G    ABOUT USING A SECOND MODEL TO GENERATE PARAMETERS

In DRUM (Sadeghian et al., 2019) and NeuralLP (Yang et al., 2017), they applied a second RNN model to generate the probabislitic logical rule parameters. This might originate from the consideration of reducing parameter space sharing useful information across similar predicates. We also compared performance with and without using a second model to generate probability weight parameters when reproducing results in DRUM. Interestingly, both settings performed more or less similarly on graph completion. For example, on all metrics on Family, UMLS and Kinship, the absolute score difference is in range $score_{\text{without}} - score_{\text{with}} \in [-0.01, +0.02]$. The only significant difference is that using a second model to generate parameters helps DRUM converge faster. This might be because the second model serves as a regularization and enables more efficient parameter-space search, but provides little performance gains.  So in our paper, we directly use learnable parameters for more elegant representation of the proposed approach.

## H    ADDITIONAL RELATED WORK

**Neural Network Models for Graph Reasoning**    Using deep neural networks for reasoning on knowledge graph and relational data is also an important direction.  There are methods that use

Logical rule: $(\nexists z_2 : sister(z_2, x)) \wedge mother(z_1, x) \wedge son(y, z_1) \Rightarrow brother(y, x)$

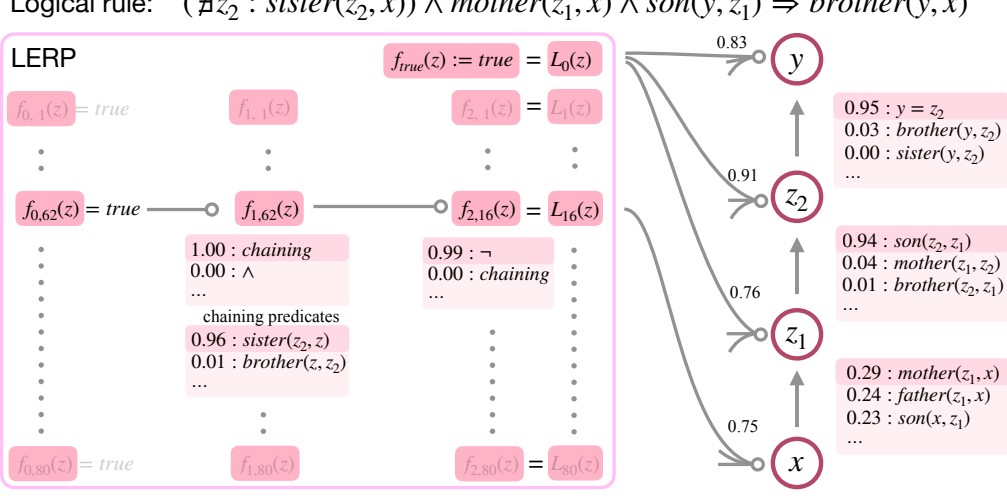

Figure 7: A diagram showing an example logical rule expressed in our model.

recurrent neural networks (RNN) such as Kaur et al. (2019) and graph neural networks (GNN) such as CoMPILE (Mai et al., 2021) ExpressGNN (Zhang et al., 2019) ConGLR (Lin et al., 2022) and GraIL (Teru et al., 2020). Our work, in contrast, focuses more on explicit modelling logical rules for better interpretability.

**Soft Logic, Probabilistic Logic and Natural Logic**  Besides logic rule learning for knowledge graph completion, there are also important work in other domains for learning complex logical rules. For example, probabilistic soft logic applies machine learning framework for developing probabilistic models (Bach et al., 2017; Srinivasan et al., 2022; Pryor et al., 2022; Hu et al., 2016; Manhaeve et al., 2018). Markov logic network (MLN) (Richardson & Domingos, 2006; Mihalkova & Mooney, 2007; Bach et al., 2017; Domingos & Lowd, 2019) is another probabilistic logic approach which applies Markov network to model first-order logic. A similarly related domain is statistical relational learning, which aims at building domain models for relational structure (Getoor et al., 2001; Koller et al., 2007; Domingos & Richardson, 2001; Nickel et al., 2011). Finally, this work is also related to neural or neural-symbolic approaches for modelling logic, such as Mao et al. (2018); Dong et al. (2018); Yi et al. (2018); Marra & Kuželka (2021); Qu & Tang (2019).

**Structure Learning of Graphical Models**  Graphical models are computational models for interdependency among random variables. Therefore, learning of probabilistic logical rules is also related to graphical model structure learning like Tsamardinos et al. (2006); Spirtes et al. (2000); Heckerman et al. (1995); Friedman et al. (2013); Hartemink et al. (2000); Tsamardinos et al. (2006); Schmidt et al. (2007); Lukashin et al. (2003).

## I  NUMBER OF PARAMETERS

The number of parameters indeed provides helpful information for comparison among models. On Family dataset for example, the total number of parameters in our approach is 46k, with 31k parameters in chains and 15k parameters in the LERP representation. On other datasets, we use 76k parameters for Kinship, 175k for UMLS, and 50k for WN18 and WN18RR. The number mainly depends on the number of predicate types. The DRUM baseline, however, uses 802k parameters on Family dataset, most of which (796k) are in the LSTM cells. DRUM uses 812k parameters for Kinship, 828k for UMLS and 807k for WN18 and WN18RR.

## J  DIAGRAM ILLUSTRATION OF LERP

In Figure 7, we plot a diagram of the networks that express an example rule. The example rule is the same rule as the first example in Table 3. At the right side of the figure, we show the main

chain and its parameters. At the left of the figure, we represent LERP and the matrix of intermediate functions. Note that in practice, we added $L_0(z) = true$ as an additional logical function in LERP. This is theoretically equivalent to the formulation in the main paper, but helps our model converge more stably. We also omit the selection edges that are not associated with maximum weight. For example, in the figure, the first variable in the chain $x$ selects logical function $L_{16}$ with weight 0.75, and we omit edges from $x$ that show other logical function selections with less weights in the diagram.

