# OpenReview forum: "Logical Entity Representation in Knowledge-Graphs for Differentiable Rule Learning"
_ICLR.cc/2023/Conference — ICLR 2023 poster_

### Official Review · Reviewer_i7dT · 2022-10-24

**Confidence:** 3
**Correctness:** 4
**Technical Novelty And Significance:** 3
**Empirical Novelty And Significance:** 3
**Recommendation:** 8

**Clarity, Quality, Novelty And Reproducibility:**

There are some nits with the presentation that I note above. The novelty in the context of knowledge graph completion is there, though the paper could benefit from more literature review in e.g. graphical models. Generally the paper is well written, and details for reproducibility are included.

**Strength And Weaknesses:**

Love the examples of logical rules discovered from the family dataset. These kind of examples are great for getting a sense of the method. Maybe in the supplementary material, a diagram of the networks that express some of those rules could be helpful to see.

The description of the model itself could due with being a little more computationally grounded. The probability / alpha / beta parameters are constrained to the simplex, which presumably is accomplished with a softmax. Being more explicit about this could be helpful.

More importantly, listing the entire # of parameters each method uses for each of the datasets would be helpful for comparison.

There is a lot of work on soft logic, it would be nice to have a bit more detailed literature review here.  While the distinction between horn clauses and the richer clauses you learn here is clear, there is certainly work outside of KG completion that learns more complex probabilistic rules, e.g. graphical model structure learning, probabilistic soft logic, natural logic, Markov logic networks, etc.

When combining with TransE, do you retrain your own TransE for the baseline for apples-to-apples? The original results in the TransE paper are quite a bit lower than what you can get with a well trained TransE using modern methods.

Nits:

- For "Embedding-Based Neural Graph Reasoning", could cite methods based on box embeddings (query2box, BoxE, etc)

- 5.1 “knowldge graph completion” should be “knowledge graph completion”

**Summary Of The Paper:**

The paper is well motivated, citing the rather broad literature on learning logical and probabilistic rules for KG completion, and proposing a method that can learn non-chain-like rules. The method makes sense. They show good results on several KG completion datasets, comparing against relevant baselines.

**Summary Of The Review:**

The paper presents a clearly motivated model for knowledge graph reasoning that solves problems with existing methods and compares against relevant baselines. The examples of learned rules are greatly appreciated, and the experimental analysis is thorough. The KG completion community would benefit from this paper’s publication.

---

> ### Author Response · Authors · 2022-11-17
> **Response to Reviewer i7dT**
>
> Thank you for your constructive comments. We are glad that you find the approach well-motivated and our examples interesting. Per your suggestions, we make these modifications in the updated version of the paper:
>
> 1. We totally agree that a diagram of the network will help illustrate the example and the model better. In the updated PDF we upload a figure in Appendix J to show how our model express an example logical rule. We hope you like the figure!
>
> 2. It is indeed highly important to make the technical details clearer. Specifically, regarding the questions you pointed out, we included these clarifications in the updated PDF:
>
>     * (a) The probability / alpha / beta parameters are indeed produced by a softmax over free parameter tensors. For example, to constrain $\alpha_{r,j,i}$ to form a distribution weights over set $r\in\mathcal{R}$, we first build free parameters $\alpha_{r,j,i}'\in\mathbb{R}$, and calculate $\alpha_{r,j,i}$ with a softmax operation $\alpha_{r,j,i}=\frac{e^{\alpha_{r,j,i}'}}{\sum_{r\in\mathcal{R}}e^{\alpha_{r,j,i}'}}$. The same also applies to $\beta$ and $p$.
>
>     * (b) We agree that discussing the number of parameters will make better comparison among models. On Family dataset for example, the total number of parameters is 46k, with 31k parameters in chains and 15k parameters in the LERP representation. The DRUM baseline, however, has a larger number of 802k parameters, most of which (796k) are weights in the LSTM cells. On other datasets, we use 76k parameters for Kinship, 175k for UMLS, and 50k for WN18 and WN18RR. The number mainly depends on the number of predicate types. This number for DRUM is 812k for Kinship, 828k for UMLS and 807k for WN18 and WN18RR. We add these details to Appendix I in the updated PDF.
>
> 3. We totally agree that comparing our approach to methods outside KG completion will better position our work in the logical rule learning area. In updated Appendix H, We add comparison with graphical model structure learning, probabilistic soft logic, natural logic, and Markov logic network, and neural-symbolic approaches for modelling logic.
>
> 4. About the experiment with TransE, in the original submission we used the training method described in the TransE paper. Under that setting we retrained all methods in our experiment (TransE, TransE+LERP and TransE+LERP+entropy) on Family dataset to ensure fair comparison among methods, and got the results in Figure 5. Enlightened by your suggestion, we also check out a relatively newer and highly cited public implementation in https://github.com/thunlp/OpenKE and apply it to Family dataset. Using that implementation, we get results more or less similar to our original numbers:
> | Model | MRR | Hits@1 | Hits@3 | Hits@10 |
> | - | - | - | - | - |
> | TransE | 0.217 | 0.035 | 0.313 | 0.606 |
> | TransE + LERP | 0.258 | 0.077 | 0.350 | 0.645 |
> | TransE + LERP + Entropy | 0.263 | 0.084 | 0.359 | 0.606 |
>
>
> 5. We also corrected the two nits that you pointed out in the updated PDF. Discussions about box embeddings are also added along to Related Work section.

---

### Official Review · Reviewer_F1BJ · 2022-10-25

**Confidence:** 4
**Clarity, Quality, Novelty And Reproducibility:** The quality of this paper is good and…
**Correctness:** 2
**Technical Novelty And Significance:** 3
**Empirical Novelty And Significance:** 2
**Recommendation:** 5

**Strength And Weaknesses:**

### Strength
1. The method can learn logical rules more complex than traditional chain rules.
2. The rules learned by this method are interesting (table 3).

### Weakness
1. The novelty seems limited considering the existing works NLIL[1], which is highly-related but missing in this paper. NLIL[1] is a related work proposing tree-like rules, which needs to be analyzed and compared. (Although the defined tree-like rules are somehow diffferent)
2. In page 5, the physical meanings of the above equations about $\mathbf{v}$ require more explanations. It the value in $\mathbf{v}$ always in (0, 1) (even after summation and multiplication)? Otherwise $1- \mathbf{v}$ would be weird.
3. As mentioned in paper, $p$, $\lambda$, etc. are learnable parameters. I wonder whether the authors have considered use some model to generate such parameters? For example, DRUM use RNN to generate the weight $\alpha$.

[1] Yang Y, Song L. Learn to Explain Efficiently via Neural Logic Inductive Learning[C]//International Conference on Learning Representations. 2019.

**Summary Of The Paper:**

This paper proposes a method for learning complex logical rules considering semantics hidden in subgraphs rather than simple chain-like rules. The authors conduct extensive experiments on both large KG datasets and relatively small statistical rule learning datasets.

**Summary Of The Review:**

This paper is somehow interesting, but considering key related works are missing, and the novelty may be weakened compared to the authors claim.

---

> ### Author Response · Authors · 2022-11-17
> **Response to Reviewer F1BJ (1/2)**
>
>
> Thank you for your insightful comments. We are glad that you find the learned rules interesting and the experiments extensive. We hope that the following explanations will help address your concerns (and we also updated the PDF to include these improvements):
>
> 1. We see that you raise a interesting question about NLIL, which we agree is a very interesting baseline and need be compared with. In the updated PDF we included additional discussions in Appendix E. A major difference between LERP and NLIL is that, LERP is able to model more complex graph structure in logical rules, and encode more contextual information about nodes in the middle of chain rules. Although NLIL also aims at going beyond chain-like rules, its definition of "tree-like rules" is indeed very different from ours (which is also mentioned in your comments). More specifically, the rules in NLIL are formulated as multiple chain-like rules combined together logically. As we will explain in the following sub-points, this formulation still overlooks contextual information for middle nodes in logical chains, and can even degrade to simple baselines like NeuralLP (Yang et al, 2019) in some settings. This problem can be viewed from the following perspectives:
>
>     * (a) **Formulation of rules**: NLIL has three parts working hierarchically. (i) The first part is responsible for searching among unary predicates $\mathcal{U}$ and binary predicates $\mathcal{B}$. (ii) The second part searches for "primitive statements", which are essentially chain-like rules with either one free variable $x$ at an end or two free variables $x$ and $x'$ at two ends. This can be seen from the definition of $\psi_k$ (Equation(9) in NLIL paper): $\psi_k(x, x')$ can be written in one of the two cases: $\sigma(\mathbf{1}^\top  \mathbf{M}^\top_k \prod_{t'=1}^T\mathbf{M}^{(t')}\mathbf{v_{x'}})$ or $\sigma(\mathbf{v_x}^\top \prod_{t=1}^T{\mathbf{M}^{(t)}}^\top \mathbf{M}^\top_k \prod_{t'=1}^T\mathbf{M}^{(t')}\mathbf{v_{x'}}).$ Both cases belong to the family of chain-like rules described in Equation 1 and 3 in our paper. (iii) The final step logically combines primitive statements, which applies logical operators $\{\land \lor \lnot\}$ over chain-like rules. The instantiation of the rules in the graph are still multiple chains spanning from $x$, or connecting $x$ and $x'$. Similarly, chain-like rule baselines DRUM (Sadeghian et al., 2019) and NeuralLP (Yang et al, 2019) also models disjunction of multiple chains. Our formulation, however, is able to model more complex graph structures (trees or branched chains).
>
>     * (b) **Example rules**: NLIL presents examples of learned logical rules in its Table 4. Its rules can still be parsed as logical combination of multiple chains. For example, the first rule for $\texttt{Person}(X)$ can be seen as logical disjunction $\lor$ combining these chains: $\texttt{chain}_1:\texttt{Shirt}(Y_1)\land\texttt{Wearing}(Y_1,X)$, $\texttt{chain}_2:\texttt{Pants}(Y_2)\land\texttt{Wearing}(Y_2,X)$ and $\texttt{chain}_3:\texttt{Street}(Y_3)\land\texttt{Wearing}(Y_3,X)$. The same also applies to other examples in the table. On the other hand, baselines DRUM and NeuralLP also model an implicit conjunction operation $\lor$ over chain-like rules. For example, one can gather DRUM Table 7 chain-like rules equivalently to a conjunction form: $\texttt{wife}(A, B) \leftarrow \texttt{husband}(B, A) \lor (\texttt{mother(A, C), \texttt{father}(B, C)}) \lor (\texttt{son}(C, A)\land\texttt{father}(B, C))$. So the examples in NLIL still falls within the discussion of multiple chain-like rules.
>
>     * (c) **Experiment results**: On knowledge graph completion experiments (Table 1 in NLIL paper), NLIL's absolute scores are still similar or inferior to chain-like rule learning baselines like NeuralLP, with gains in MRR and Hits of no more than 0.01 absolute value. This also partially validates that NLIL's expressiveness in graph reasoning domain is similar to chain-like rules learning.
>
> In contrast, our proposed LERP represents more complex logical rules than chain-like rules. It can instantiate on trees or branched chains in the knowledge graph and obtains substantially higher scores compared to chain-like rule learning baselines.
>
> [1] Sadeghian, Ali, et al. "Drum: End-to-end differentiable rule mining on knowledge graphs." Advances in Neural Information Processing Systems 32 (2019).
>
> [2] Yang, Fan, Zhilin Yang, and William W. Cohen. "Differentiable learning of logical rules for knowledge base reasoning." Advances in neural information processing systems 30 (2017).
>
> [3] Yang, Yuan, and Le Song. "Learn to Explain Efficiently via Neural Logic Inductive Learning." International Conference on Learning Representations. 2019.

---

> ### Author Response · Authors · 2022-11-17
> **Response to Reviewer F1BJ (2/2)**
>
> 2. We totally agree our draft can be improved by fixing some technical details. Related to your comment, we fix and clarify the following points here (and also include them in the updated PDF):
>
>     * (a) After the chaining operation, we actually added a soft clamping function $\text{clamp}(x) = 1-\exp(-x)$ element-wise, to guarantee that values are within $[0, 1]$. So the calculation of chaining operator should have been: $v_{j,i;\text{chaining}}=\text{clamp}(v^\top_{j-1, I}  \sum_{r\in\mathcal{R}} \alpha_{r,j,i} A_r)^\top$.
>
>     * (b) Having the fix above, in Section 4.1 all $\mathbf{v}$ are naturally guaranteed to be in range $[0, 1]$. This is because the parameters $\alpha$ $\beta$, and $p$ are limited to the probability simplex. It is easy to see that, recursively speaking, when all previously calculated $\mathbf{v}$ are in range $[0, 1]$, any newly calculated $\mathbf{v}$ is also in $[0, 1]$.
>
>
> 3. We notice that you raised a very interesting concern about the generation of parameters. We also compared performance with and without using a second model to generate probability weight parameters when reproducing results in DRUM. Interestingly, both settings performed more or less similarly on graph completion. For example, on all metrics on Family, UMLS and Kinship, the absolute score difference $score_{\text{without}} - score_{\text{with}} $  is in range $ [-0.01, +0.02]$. The only significant difference is that, using a second model to generate parameters helps DRUM converge faster. This might be because the second model serves as a regularization and enables more stable parameter-space search. So in our paper, we directly use learnable parameters for cleaner mathematical representation of the proposed approach. We include this discussion in Appendix G in the updated PDF.
>
> [1] Sadeghian, Ali, et al. "Drum: End-to-end differentiable rule mining on knowledge graphs." Advances in Neural Information Processing Systems 32 (2019).

---

### Official Review · Reviewer_N1P8 · 2022-10-27

**Confidence:** 3
**Correctness:** 3
**Technical Novelty And Significance:** 2
**Empirical Novelty And Significance:** 2
**Recommendation:** 6

**Clarity, Quality, Novelty And Reproducibility:**

Clarity: Good
Novelty: minor contribution
Reproducibility: very good

**Strength And Weaknesses:**

Pros:
+ It explores a new alternative to learn probabilistic logical rules
+ Paper is well written and organized

Cons:
- Related work on logic rule learning does not cite many important
references about the main method for learning first order rules:
inductive logic programming. The same happens with the related work on
Probabilistic logical rule learning where authors omit refs to
ProbLog, CLPB(n), SlipCover, ProbFoil, SKiLL, among others.
There are also several works that are neurosymbolic like RNN, and
others which try to capture the neuronal network structure from the
relations in the data (e.g., Kaur et al. and Sourek et al).


**Summary Of The Paper:**

This paper proposes LERP (Logical Entity RePresentation), a model that
uses logical rule learning, but which embeds information about the
objects represented by the logical variables in the form of a vector
of logical functions.

**Summary Of The Review:**

This paper proposes LERP (Logical Entity RePresentation), a model that
uses logical rule learning, but which embeds information about the
objects represented by the logical variables in the form of a vector
of logical functions.

Related work on logic rule learning does not cite many important
references about the main method for learning first order rules:
inductive logic programming. The same happens with the related work on
Probabilistic logical rule learning where authors omit refs to
ProbLog, CLPB(n), SlipCover, ProbFoil, SKiLL, among others.

There are also several works that are neurosymbolic like RNN, and
others which try to capture the neuronal network structure from the
relations in the data (e.g., Kaur et al. and Sourek et al).

The work looks quite interesting as it presents an alternative way and
optimization to find interpretable models. Just out of curiosity,
authors say that RNNLogic uses 100 to 200 rules and DRUM uses 1 to
4. I don't know if I understood that well. ILP systems such as Aleph
or Foil can learn thousands or millions of rules in a decente amount
of time. How much time does your method take to learn the
probabilistic logical rules? The search for subgraphs for entities
(logical variables) is naturally implemented in ILP systems. It would
be interesting to compare the probabilistic logical rule learning
method used in this work with a probabilistic ILP system like, for
example, ProbFoil.

---

> ### Author Response · Authors · 2022-11-17
> **Response to Reviewer N1P8**
>
> Thank you for the helpful comments. We are very happy that you find our submission well written and the work interesting. With respect to your questions and concerns we prepare the following responses:
>
> 1. We totally agree that the submission can be improved by including discussions about a broader set of prior work, like the ILP field and more traditional logic rule learning methods. We updated the reference in the Related Work section and in Appendix H.
>
> 2. We see you raised an interesting question about the rule learning speed, especially compared with traditional ILP methods.
> We respond to your question from the following two aspects (which is also added to Appendix F):
>
>     * (a) As far as we can see, differentiable rule learning methods for knowledge graph completion (including our approach as well as RNNLogic[2] and DRUM[1]) normally do not set the number of logical rules to very high. A typical number usually ranges from 3 or 4 to a few hundreds of rules. A main reason is that these work focus more on downstream evaluation metrics, and there are quickly diminishing returns when increasing the number of rules beyond that (see Figure 4(c) in our submission).
>
>     * (b) On Family dataset for example, our model trains for 2.8 minutes and gets 4800 rules (200 rules / target predicate $\times$ 24 target queries). In comparison, DRUM gets the same number of rules in 1.2 minutes, while RNNLogic uses 10 minutes.
>
>
> [1] Sadeghian, Ali, et al. "Drum: End-to-end differentiable rule mining on knowledge graphs." Advances in Neural Information Processing Systems 32 (2019).
>
> [2] Qu, Meng, et al. "RNNLogic: Learning Logic Rules for Reasoning on Knowledge Graphs." International Conference on Learning Representations. 2020.

---

### Official Review · Reviewer_dYXp · 2022-10-27

**Confidence:** 4
**Correctness:** 2
**Technical Novelty And Significance:** 3
**Empirical Novelty And Significance:** 3
**Recommendation:** 5

**Clarity, Quality, Novelty And Reproducibility:**

### Novelty

This work approaches an important problem in ILP, which is learning more expressive rules from the KGs. The authors propose to extend the standard backward-chaining methods, which only learn chain-like rules, by also learning the "branches" rooted from the intermediate entities in the main path. While this family of rules is not as general as "local subgraphs" as has been claimed by the authors, it is already an interesting extension.


### Quality

Due to presentation issues, I'm unable to assess the proposed method in detail, but the general methodology is sensible and computationally feasible. Some of my concerns are as follows:

The complexity of adjacency matrix multiplication should be O(Kn^3) instead of O(Kn^2)

If my understanding is correct (see clarity), while one can set depth $T$ and control the complexity of the functions in Fig (2), the number of hops permitted for each $L(z_i)$ is always set to 1. This means the LERP effectively learns a "branched" chain-like path, where the "main" path is of length K, and the branches of length 1. This is rather limited to be considered as "local subgraphs". Nevertheless, it is still a nice extension to the original chain-like ones, but this aspect should be stated clearly in the paper.


The 6 types of functions introduced on page 5 require more justifications as they are not very intuitive. For example, it is unclear what $v_{i,j}$ represents: judging from the chaining function, $v_{i,j}$ seems to be the random walk feature vector, but I'm not sure why we set $v_{i,j}$ to be 1 for the first column. Also, if $v_{i,j}$ is the random walk vector that contains counts of unique paths to the entities then why one should perform conjunction and disjunction on it?


### Clarity

The presentation, especially section 4, should be largely improved. Right now

I'm confused about the notion of "columns" referred to in Fig 2 and section 4.1.
- Why does one need a matrix of $f_{i,j}$? How is it related to the "tree-like" function in Def 1?
- The ranges of $i, j$ are confusing. I suppose they are $m$ and $T$? If so, how do you pick values for $m,T$ and why?
- It is unclear why the first columns are set to true; what do the entries of the first column represent? Is there any graphical interpretation of Fig 2? Or in other words, what subgraph does Fig 3 represent?
- Does the second column always use the chaining function?



Notation issues:
- $w_i$ used in Def 1 without definition. How is it different from $z_{i,j}$?
- What is $T$ in "There are a total of T+ 1columns of intermediate functions"?


**Strength And Weaknesses:**

Strength
- Approaches an important problem of learning more expressive FOL rules with ILP
- LERP learns a more expressive family of rules than prior ILP methods.

Weaknesses
- Presentation should be largely improved. The current draft, especially section 4, is difficult to follow and misses some important technical details.
- The learnable rule family seems to be still far from being considered as local subgraphs.

**Summary Of The Paper:**

The paper proposes LERP, a differentiable ILP method that mines FOL rules from knowledge graphs. Compared to the prior backward-chaining methods which only learn chain-like paths in the graph, the authors propose to learn more expressive family of rules by considering the local subgraphs along the main path. In the experiment, LERP is evaluated with several ILP and graph embedding methods and shows better performance and interpretability.

**Summary Of The Review:**

While this work presents an interesting extension to the existing ILP methods, there are some serious presentation issues to be fixed. At this stage, I cannot recommend for acceptance but I will be happy to raise my score if the authors could address my concerns and I'll be looking forward to reading the revised draft.

---

> ### Author Response · Authors · 2022-11-16
> **Response to Reviewer dYXp (1/2)**
>
> Thank you for the detailed and constructive comments. We are glad that you find the approached problem important and the extension interesting. Regarding your mentioned concerns and questions, we improved the draft to include the following discussions:
>
> 1. We see that you raised an interesting concern regarding the computational complexity of adjacency matrix multiplication.
> In our work, we adopt a trick that, by properly selecting multiplication order in our problem, the complexity can be reduced. We notice that the formula is in the form $\mathbf{v}^\top_x\prod^K_{k=1} A_{r_k}$.
> In calculation we choose to first multiply $\mathbf{v}^\top_x$ with $A_{r_1}$. and then multiply $A_{r_2}$, so on and so forth. In this way, each step we are only calculating a vector-matrix multiplication, which is $O(n^2)$ complexity. So in total $K$ vector-matrix multiplications need $O(Kn^2)$ complexity.
> The same trick is also applied to the formula in Section 4.2.
> We add this clarification in the updated PDF to make the description clearer.
>
> 2. We understand the reviewer has a concern on the expressiveness of LERP, and we've taken steps to address it in our new revision. We explain that, by setting depth $T$, the number of hops in $L(z)$ can be larger than 1. For example, in Figure 2(a), if we change $f_{3,1}$ to be a chaining function over $f_{2,1}$, then $L(z)$ can perform 2-hop detection/instantiation. In that case, $L(z)=\exists z_3: R_3(z_3, z)\land(\exists z_1: R_1(z_1, z_3) \land \exists z_2: R_2(z_2, z_3))$. Specifically, it detects if $z$ connects an entity with $z_3$ with $R_3$, while $z_3$ itself connects to $z_1$ and $z_2$ with $R_1, R_2$. If $T$ is larger, the number of hops in $L(z)$ can be larger. In this sense, the rules learned in this work are more similar to "complexly branched" chain-like paths, where each branch itself can have its own branch(es).
> As a concrete example from Table 5, the bottom-right function $\exists z_1,z_2:\text{mother}(z_1,e)\land\text{niece}(z_1,z_2)$ is such a two-hop logical function.
>
>
>     The reviewer also raised an interesting question about modelling general local subgraphs. Indeed in this paper, we limit the type of local subgraphs to trees, so as to guarantee the computational efficiency ($O(Cn^2)$) as described in the draft. In fact, when the local subgraphs are more complex than trees (i.e., containing loops), the computational complexity to instantiate them can be larger. According to a popular conjecture in the fine-grained complexity area [1], to detect the existence of a triangle in a dense graph, any algorithm requires $\Omega(n^{\omega-o(1)})$ time, where $\omega$ is the matrix multiplication exponent with current best bound around 2.37. Here a triangle constitutes a simplest violation to the tree constraint. We can easily check that the following logical rule requires the above computational complexity to determine: $H(x,y) \Leftarrow (x \neq z) \land (\exists z_1, z_2: r(z, z_1)\land r(z_1, z_2)\land r(z_2, z))  \land (z \neq y)$ where $r$ is an arbitrary relation. The reason is that, the logical function for $z$ induces an problem equivalent to detecting a triangle in the graph. This is why we choose to constrain ourselves to tree-like logical functions in this paper.
>
>
> 3. We see that the reviewer raises a question about the meaning of $\mathbf{v}$ vectors. We first add a fix to the chaining function in the updated PDF: we had actually applied a soft clamping function $\text{clamp}(x) = 1-\exp(-x)$ element-wise to $v_{j,i; \text{chaining}}$, to guarantee that values are within $[0, 1]$. So the calculation of chaining operator should have been:
> $v_{j,i;\text{chaining}}=\text{clamp}(v^\top_{j-1, I}  \sum_{r\in\mathcal{R}} \alpha_{r,j,i} A_r)^\top$.
> Having added this fix, generally speaking, all vectors $\mathbf{v}$ are soft-logic vectors taking values from range $[0, 1]$. They approximately represent the values of the probabilistic logic rule on the entity set. The disjunction and conjunction functions then follow the meaning of $\mathbf{v}$ in this sense. The reason for setting $\mathbf{v}$ to be an all-1 vector in the first column is that it represents an all-true function. Then, the chaining function in the second column represents $f_{1, i}(z) = \exists z_1: R_1(z_1, z)$, indicating if there are ``any'' $z_1$ connecting to $z$ with $R_1$, while $z_1$ can be arbitrary. We also clarified this part in the updated draft.
>
>
> [1] Abboud, Amir, and Virginia Vassilevska Williams. "Popular conjectures imply strong lower bounds for dynamic problems." 2014 IEEE 55th Annual Symposium on Foundations of Computer Science. IEEE, 2014.

---

> ### Author Response · Authors · 2022-11-17
> **Response to Reviewer dYXp (2/2)**
>
> 4. We appreciate your callouts to clarify some technical points to make the representation easier to understand. We revised the presentation in the PDF with the following improvements according to your comments:
>
>     * (a) **Why we need a matrix of $f$ functions**: the motivation of LERP is to represent a family of tree-like logical functions defined in Definition 1. Taking Figure 2(a) as an example, we observe that this $L(z)$ can be seen as constructed by a hierarchy of intermediate functions $f_{1,1}(z)=\exists z_1: R_1(z_1, z)$, $f_{1,2}(z)=\exists z_2: R_2(z_2, z),~f_{2,1}(z)=(\exists z_1: R_1(z_1, z) \land \exists z_2: R_2(z_2, z)), \cdots$, etc. In Figure 2(a), $L(z)$ can be built within a depth-3 hierarchy with 4 leaf-functions. In this work we aim at learning a list of logical functions $L_1, L_2, \cdots$, but we do not know beforehand the exact structure of each logical function. So we propose to design with a ``redundancy'' principle, by forming a matrix of intermediate functions. A well learned matrix can ideally contain a desired hierarchy within it.  In Figure 2(b) for example, $f_{3, 1}^\Theta$ is probabilistically representing the same logical form as $f_{3,1}$ in Figure 2(a).
>
>     * (b) **About the range of $i, j$**: The matrix of functions has $m$ rows and $T+1$ columns. $i$ is in range $\\{1, \cdots, m\\}$ and $j$ is in range $\\{0, 1, \cdots, T\\}$. $m$ and $T$ are hyper-parameters, and we search for a combination of $m$ and $T$ according to validation set performance. The searching range is $m\in\\{3, 10, 40, 80, 200, 400\\}$ and $T\in\\{0,1,2,3,4\\}$. $T=0$ is equivalent to not using LERP and our model reduces to a similar formulation as DRUM. This grid search leads us to adopt $m=80$ and $T=2$ for LERP.
>
>     * (c) **About the meaning of the first column and Fig 3**: the first column is a list of all-true functions $f_{0, i}(z) = true$. This accords with Definition 1, where the construction of the function family always starts from a true function. The score for $f_{0, i}$ is an all-1 vector. Following the logical interpretation of Fig 2 in (2.a) above, its graphical interpretation can be viewed as follows. This logical function $L(z)=f_{3,1}(z)$ detects if there are two (not necessarily distinct) entities $z_1,z_2$ linked to $z$ with $R_1$ and $R_2$ respectively, or, if an entity $z_3$ links to $z$ with $R_3$, or if an entity $z_4$ links to $z$ with $R_4$. Figure 3, on the other hand, shows the PCA of the whole LERP vectors of entities. So each dimension in Figure 3 is a principal component in the LERP space, and is thus a mixture over multiple LERP functions.
>
>     * (d) **Whether the second column always use the chaining function**: in our code implementation, we always set the second column $f_{1, \cdot}$ to only use chaining function. This is because, although Definition 1 does not limit which operations use, only a chaining function produces meaningful logical functions in the second column. A chaining function in the second column outputs functions in the form of $\exists z_1: R_1(z_1, z)$. A $\land$-merging or $\lor$-merging operation, however, still outputs an all-true function. A negation operation will output an all-false function. Therefore, not using $\land$-merging, $\lor$-merging and $\lnot$ in the second column does not affect the expressiveness of LERP.
>
>     * (e) **The notation of $w$ and $z$**: both notations are variables of entities, and take values from entity set $\mathcal{E}$. They can be actually used interchangeably. In the updated PDF we clarified this part.
>
>
>     * (f) **What $T$ is in "There are a total of $T+1$ columns of intermediate functions"**: in the matrix of intermediate functions $f$, the index of the function column $j$ ranges within $\\{0, 1, \cdots, T\\}$. This is because we also include the \textit{true} functions as the first column of functions $f_{0, i}$.

---

### Author Response · Authors · 2022-11-18
**General Response**

Dear AC and reviewers,

We thank all reviewers for their helpful comments! We are happy to see reviewers recognize that the tackled problem important and our proposed extension interesting. We also greatly appreciate that reviewers find the learned logical rules more complex and interesting.

We responded to questions and concerns in each individual response, and also improved the submission with revisions according to your helpful comments and suggestions. In summary, we addressed the following concerns:

1. **Presentation** We updated the Section 4 to clarify a few technical concerns (reviewer dYXp and i7dT);

2. **Related work** We included more related work to make discussion more comprehensive in Section 2 and Appendix H (N1P8, F1BJ and i7dT), and clarified the distinction between our approach and NLIL in Appendix E (F1BJ);

3. **Model details** We included more details like the number of parameters in Appendix I (i7dT) and rule learning speed in Appendix F (N1P8);

4. **Illustration diagram** We added a diagram in Appendix J to better illustrate the model representing example rules (i7dT).

We are more than happy to discuss about any further questions that you might have!

---

### Author Response · Authors · 2022-12-02
**Looking forward to further discussions!**

Dear all reviewers,

In discussion stage 1, we revised and improved the paper based on your constructive and helpful comments. We addressed concerns in presentation and references, and added more discussions about comparison to previous work, model performance and implementation, and a new illustration diagram to visualize the model and examples. We would sincerely appreciate it if you could check the responses and re-evaluate our paper.

We now look forward to having more discussions with you about any further questions and suggestions you might have in discussion stage 2. Again, we are really grateful for all your efforts in reviewing our paper!

---

### Decision · Program_Chairs · 2023-01-20

**Decision:**

Accept: poster

**Justification For Why Not Higher Score:**

While the ideas in the paper are interesting and useful, I am not sure to what extent the reviewers were impressed by them to consider this paper as a spotlight or an oral.

**Justification For Why Not Lower Score:**

The paper stands on solid ground.

**Metareview: Summary, Strengths And Weaknesses:**

This paper proposes a way to learn the rules for inference from knowledge graphs. The paper overall presents interesting ideas, and some of the reviewers were excited by it. The authors did more than expected to fix the paper according to the reviewers' comments. The empirical results are also relatively strong, which is an important strength in this case.

**Note From Pc:**

if the above contains the word "oral" or "spotlight" please see: "oral" presentation means -> notable-top-5% and "spotlight" means -> notable-top-25%. As stated in our emails, we are disassociating presentation type from AC recommendations

**Summary Of Ac-Reviewer Meeting:**

There was no need for a meeting to make an informed decision.